# In Vitro Biological Evaluation of Aprepitant Based ^177^Lu-Radioconjugates

**DOI:** 10.3390/pharmaceutics14030607

**Published:** 2022-03-10

**Authors:** Paweł K. Halik, Przemysław Koźmiński, Joanna Matalińska, Piotr F. J. Lipiński, Aleksandra Misicka, Ewa Gniazdowska

**Affiliations:** 1Centre of Radiochemistry and Nuclear Chemistry, Institute of Nuclear Chemistry and Technology, 03-195 Warsaw, Poland; p.kozminski@ichtj.waw.pl (P.K.); e.gniazdowska@ichtj.waw.pl (E.G.); 2Department of Neuropeptides, Mossakowski Medical Research Institute, Polish Academy of Sciences, 02-106 Warsaw, Poland; jmatalinska@imdik.pan.pl (J.M.); plipinski@imdik.pan.pl (P.F.J.L.); misicka@chem.uw.edu.pl (A.M.); 3Faculty of Chemistry, University of Warsaw, 02-093 Warsaw, Poland

**Keywords:** aprepitant, radiopharmaceuticals, neurokinin 1 receptor antagonist

## Abstract

Currently, the search for promising NK1R-positive tumor-targeting radiopharmaceuticals based on the structure of small molecular antagonists of neurokinin-1 receptor can be observed. Following this trend, we continued our evaluation of aprepitant-based ^177^Lu-radioconjugates in terms of future oncological applications. For this purpose, three novel aprepitant homologues were synthesized to broaden the previously obtained derivative portfolio, functionalized with the DOTA chelator and labeled with ^68^Ga and ^177^Lu. The newly evaluated radioconjugates showed the intended significant increase in lipophilicity compared to the previous ones, while maintaining stability in the human serum. Then, in a receptor binding study to the human NK1 receptor, we compared the two series of ^177^Lu-radioconjugates of aprepitant with each other and with the reference Substance P derivative currently used in glioblastoma therapy, clearly indicating the high affinity and better binding capacity of the novel radioconjugates. The in vitro experimental results included in the presented study, supported by labeling optimization, radioconjugate characterization and docking modeling of new aprepitant-derived radioagents, confirm our assumptions about the usefulness of aprepitant as a NK1R targeting vector and point out the perspectives for the forthcoming first in vivo trials.

## 1. Introduction

Over the last few years, a significant increase of the interest in small molecular antagonists of neurokinin-1 receptor (NK1R) in terms of oncological applications can be observed. Some reports indicate new perspectives for the use of aprepitant [1] and of other NK1R antagonists [2,3,4] as vectors for selective radiopharmaceutical agents for NK1R-positive tumors.

At the same time, it results in a distraction from the radiopharmaceuticals based on derivatives of Substance P (SP) or other peptide ligands of this receptor, due to several inconveniences that characterize this group of compounds. These radiopharmaceuticals currently used in therapy of gliomas (e.g., [^213^Bi]Bi-DOTA-[Thi^8^,Met(O_2_)^11^]Substance P [5,6,7]) are administered intracavitarily in a controlled manner through an appropriate preinstalled canal port into the postoperative cavity for radiopharmaceutical administration. Unfortunately, simple intravenous administration of SP derivatives is ineffective due to their low metabolic stability, unfavorable pharmacokinetic [8], and the fact that in micromolar amounts both Substance P and its analogues could produce a severe hypotension in peripheral circulatory system [9]. Nevertheless, targeted alpha therapy utilizing [^213^Bi]Bi/[^225^Ac]Ac-DOTA-[Thi^8^,Met(O_2_)^11^]SP in locoregional application is a promising and favorable step towards the future treatment. Still, the clinicians point out that new treatment options are urgently needed [6].

Increased NK1R expression on various types of cancer correlates with tumor progression, migration and angiogenesis [10,11,12,13]. This applies to nervous system cancers as gliomas [14,15,16], but also to neoplasms of other organs. On the other hand, normal tissues show significantly lower (often negligible) levels of NK1R expression [13,16,17,18]. This fact gives the green light to the concept of selective imaging and targeting of tumor cells overexpressing NK1R.

Aprepitant may be the solution to the search for an appropriate vector to be used in targeted radionuclide tumor therapy. This representative of NK1R antagonists is of increasing attentiveness due to its broad anti-tumor activity of very promising therapeutic significance [12,13,19,20] and many years of experience in clinical application against chemotherapy-induced nausea and vomiting in oncological patients. Most significantly, it is a drug of known pharmacokinetics, metabolically stable, showing the ability to cross the blood–brain barrier after intravenous or oral administration [20,21,22]. In comparison to peptide antagonists, aprepitant is a smaller molecule with undoubtedly higher lipophilicity and a favorable biding mode towards the cognate receptor [23].

In the preceding report [1], we presented a drug design concept for convenient modification of the aprepitant structure including an extensive rationale for the selection of its triazolinone ring as a site of functionalization. We specified that this fragment of the molecule tolerates some modification, and as we indicated by several examples, it is possible to introduce modifications effectively without significant compromise to the receptor affinity. This particular knowledge of how to constructively modify the structure of the compound without adversely affecting the receptor binding affinity was implemented to report this consideration on aprepitant-based radiopharmaceuticals for NK1R-positive tumors. At the same time, we wanted to broaden previously obtained derivative portfolio with sufficiently high lipophilic derivatives and to explore the impact of much longer linkers on the chelator placement in (or nearby) the binding site, and thus on the receptor affinity of the aprepitant radioconjugate.

In this part of our research, we set ourselves the goal of obtaining three novel aprepitant-related homologues with higher lipophilicity using longer aliphatic linkers. Then, we obtained DOTA conjugates and optimized the conjugate labeling with ^68^Ga and ^177^Lu isotopes. Finally, we investigated a binding profile towards human NK1 receptor for all eight reported (synthesized now and previously) ^177^Lu-radioconjugates at their highest specific activity in reference to binding characteristics of ^177^Lu-labeled Substance P derivative currently used in clinical therapy. These results were also supplemented with the physicochemical evaluation of newly obtained radioconjugates and molecular modelling rationalization of the obtained affinity data.

## 2. Materials and Methods

Aprepitant (Santa Cruz Biotechnology Inc., Dallas, TX, USA), DOTA-NHS ester (1,4,7,10-tetraazacyclododecane-1,4,7,10-tetraacetic acid mono-*N*-hydroxysuccinimide ester) (CheMatech, France), and other substances and solvents (Sigma Aldrich/Merck, Darmstadt, Germany) were commercially available, defined as reagent grade, and applied without further purification. [^68^Ga]GaCl_3_ was eluted from the commercially available ^68^Ge/^68^Ga generator (Eckert & Ziegler, Berlin, Germany) by semi-automated syringe pump to fractionate the highest radionuclide content solution for the labeling reactions; no other processing was applied. [^177^Lu]LuCl_3_ solution in 0.04 M HCl was purchased from Radioisotope Centre POLATOM, National Centre for Nuclear Research, Otwock-Świerk, Poland. Human serum was a gift from the Regional Centre for Blood Donation and Blood Treatment in Warsaw, Poland. The HPLC analyses were performed using a semi-preparative Phenomenex Jupiter Proteo column, 4 μm, 90 Å, 250 × 10 mm, with UV/Vis (wavelength 220 nm) or/and radio γ-detection at gradient elution: 0–20 min 20 to 80% solvent B; 20–30 min 80% solvent B; total flow 2 mL/min.; solvent A: 0.1% (*v*/*v*) trifluoroacetic acid (TFA) in water; and solvent B: 0.1% (*v*/*v*) TFA in acetonitrile. Mass spectra were measured on a Bruker 3000 Esquire mass spectrometer equipped with electrospray ionization (ESI) (Bruker, Billerica, MA, USA). The NMR measurements were done in CD_3_OD or CD_3_CN on Varian-Agilent 600 MHz VNMRS spectrometer at ambient temperature, with trimethylsilane as the internal standard for chemical shifts.

### 2.1. Syntheses and Characterization of Aprepitant-Based Conjugates

#### 2.1.1. General Procedure of Syntheses of Aprepitant Derivatives with Alkyl Linker

The corresponding *N*-(terminal-bromoalkyl)phthalimide (2 equiv.) was added into the mixture of aprepitant (1 equiv.) and sodium carbonate (1 equiv.) in 100–200 µL of DMF. The reaction mixture was vigorously stirred at about 80 °C for 72 h. Then, the hydrazine monohydrate (3 equiv.) was added into the reaction mixture and left for additional 2–3 h. The progress of the reaction was monitored by HPLC. After this time, the reaction mixture was evaporated, dissolved in the HPLC mobile phase and purified by the HPLC method. Each main product was isolated and verified by MS analysis confirmation as a mono-substituted APT-alkylamine derivative. The purified products were collected, lyophilized and characterized by ^1^H-NMR analysis.

**2-(6-aminohexyl)-5-([(2*R*,3*S*)-2-((*R*)-1-[3,5-bis(trifluoromethyl)phenyl]ethoxy)-3-(4-fluorophenyl)morpholino]methyl)-1*H*-1,2,4-triazol-3-one, APT-Hex-NH_2_,** white powder, yield 24.0%, purity > 97% (HPLC-UV; t_R_ = 19.1 min), ESI-MS: calculated monoisotopic mass for C_29_H_34_F_7_N_5_O_3_: 633.60; found: 634.22 *m*/*z* [M+H]^+^; ^1^H NMR (600 MHz, CD_3_OD) (s—singlet, d—doublet, t—triplet, q—quartet, td—triplet of doublets, m—multiplet) δ (ppm), 7.73 (s, 1H), 7.56 (s, 2H), 7.39 (s, 2H), 7.12–7.09 (t, *J* = 8.8 Hz, 2H), 5.01–4.98 (q, *J* = 6.6 Hz, 1H), 4.37 (d, *J* = 2.9 Hz, 1H), 4.28–4.24 (td, *J* = 11.8, 2.5 Hz, 1H), 3.87–3.82 (m, 1H), 3.69–3.65 (m, 2H), 3.58–3.56 (d, *J* = 14.0 Hz, 2H), 3.49–3.48 (d, *J* = 2.9 Hz, 1H), 2.94–2.88 (m, 2H), 2.83–2.81 (d, *J* = 11.7 Hz, 1H), 2.52–2.47 (td, *J* = 12.0, 3.5 Hz, 1H), 1.75–1.65 (m, 4H), 1.50–1.48 (d, *J* = 6.6 Hz, 3H), 1.47–1.39 (m, 4H). In deuterated methanol, it was not possible to observe signals from the hydrogens of the amino group -NH_2_ (around 6.9 ppm) and hydrogen of the nitrogen in the triazolinone ring (around 10.1 ppm).

**2-(8-aminooctyl)-5-([(2*R*,3*S*)-2-((*R*)-1-[3,5-bis(trifluoromethyl)phenyl]ethoxy)-3-(4-fluorophenyl)morpholino]methyl)-1*H*-1,2,4-triazol-3-one, APT-Oct-NH_2_,** white powder, yield 19.1%, purity > 97% (HPLC-UV; t_R_ = 20.0 min), ESI-MS: calculated monoisotopic mass for C_31_H_38_F_7_N_5_O_3_: 661.65; found: 662.21 *m*/*z* [M+H]^+^; ^1^H NMR (600 MHz, CD_3_CN) (s—singlet, d—doublet, t—triplet, q—quartet, td—triplet of doublets, m—multiplet) δ (ppm), 10.07 (s, 1H), 7.78 (s, 1H), 7.54 (s, 2H), 7.40 (s, 2H), 7.12–7.09 (t, *J* = 8.9 Hz, 2H), 7.01 (s, 2H), 4.95–4.92 (q, *J* = 6.5 Hz, 1H), 4.38 (d, *J* = 2.8 Hz, 1H), 4.23–4.18 (td, *J* = 11.7, 2.5 Hz, 1H), 3.74–3.69 (m, 1H), 3.66–3.63 (m, 1H), 3.57–3.52 (m, 1H), 3.52–3.50 (d, *J* = 14.0 Hz, 2H), 3.47–3.46 (d, *J* = 2.8 Hz, 1H), 2.82 (s, 2H), 2.46–2.41 (td, *J* = 11.9, 3.5 Hz, 1H), 1.69–1.62 (m, 4H), 1.47–1.46 (d, *J* = 6.6 Hz, 3H), 1.34 (s, 8H).

**2-(10-aminodecyl)-5-([(2*R*,3*S*)-2-((*R*)-1-[3,5-bis(trifluoromethyl)phenyl]ethoxy)-3-(4-fluorophenyl)morpholino]methyl)-1*H*-1,2,4-triazol-3-one, APT-Dec-NH_2_,** white powder, yield 18.9%, purity > 97% (HPLC-UV; t_R_ = 20.5 min), ESI-MS: calculated monoisotopic mass for C_33_H_42_F_7_N_5_O_3_: 689.71; found: 690.32 *m*/*z* [M+H]^+^; ^1^H NMR (600 MHz, CD_3_CN) (s—singlet, d—doublet, t—triplet, q—quartet, td—triplet of doublets, m—multiplet) δ (ppm), 10.11 (s, 1H), 7.78 (s, 1H), 7.55 (s, 2H), 7.41 (s, 2H), 7.12-7.09 (t, *J* = 8.9 Hz, 2H), 6.88 (s, 2H), 4.9–4.92 (q, *J* = 6.5 Hz, 1H), 4.39 (d, *J* = 2.8 Hz, 1H), 4.24–4.19 (td, *J* = 11.8, 2.5 Hz, 1H), 3.72–3.68 (m, 1H), 3.66–3.63 (m, 1H), 3.56–3.51 (d, *J* = 14.0 Hz, 2H), 3.55–3.48 (d, *J* = 2.8 Hz, 1H), 2.95–2.91 (m, 2H), 2.87–2.85 (d, *J* = 13.9 Hz, 2H), 2.49–2.44 (td, *J* = 11.9, 3.6 Hz, 1H), 1.64–1.60 (m, 4H), 1.47 (d, *J* = 6.6 Hz, 3H), 1.37–1.30 (m, 12H).

#### 2.1.2. General Procedure of Syntheses of Aprepitant Conjugates with DOTA

Each obtained aprepitant-alkylamine derivative (1 equiv.) and the DOTA-NHS ester (1.2 equiv.) were dissolved in 100 µL of DMF, purged from oxygen with technical nitrogen and supplemented with triethylamine (3 equiv.). The reaction mixture was vigorously stirred overnight at about 50 °C. The progress of the reaction was monitored by HPLC. Afterwards, the reaction mixture was evaporated, dissolved in the HPLC mobile phase and purified by the HPLC method. Each main product was isolated and verified by MS analysis as a desired DOTA conjugate. The purified products were collected and lyophilized. Due to low, sub-microgram scale of reactions, NMR analyses were not possible to be performed.

**2-(6-(2-(1,4,7-tris(carboxymethyl)-1,4,7,10-tetraazacyclododecan-10-yl)acetamido)hexyl)-5-([(2*R*,3*S*)-2-((*R*)-1-[3,5-bis(trifluoromethyl)phenyl]ethoxy)-3-(4-fluorophenyl)morpholino]methyl)-1*H*-1,2,4-triazol-3-one, APT-Hex-DOTA,** white powder, yield 90.6%, purity > 97% (HPLC-UV; t_R_ = 17.1 min), ESI-MS: calculated monoisotopic mass for C_45_H_60_F_7_N_9_O_10_: 1019.44; found: 1020.39 *m*/*z* [M+H]^+^;

**2-(8-(2-(1,4,7-tris(carboxymethyl)-1,4,7,10-tetraazacyclododecan-10-yl)acetamido)octyl)-5-([(2*R*,3*S*)-2-((*R*)-1-[3,5-bis(trifluoromethyl)phenyl]ethoxy)-3-(4-fluorophenyl)morpholino]methyl)-1*H*-1,2,4-triazol-3-one, APT-Oct-DOTA,** white powder, yield 89.1%, purity > 97% (HPLC-UV; t_R_ = 17.7 min), ESI-MS: calculated monoisotopic mass for C_47_H_64_F_7_N_9_O_10_: 1047.47; found: 1048.43 *m*/*z* [M+H]^+^;

**2-(10-(2-(1,4,7-tris(carboxymethyl)-1,4,7,10-tetraazacyclododecan-10-yl)acetamido)decyl)-5-([(2*R*,3*S*)-2-((*R*)-1-[3,5-bis(trifluoromethyl)phenyl]ethoxy)-3-(4-fluorophenyl)morpholino]methyl)-1*H*-1,2,4-triazol-3-one, APT-Dec-DOTA,** white powder, yield 94.5%, purity > 97% (HPLC-UV; t_R_ = 18.3 min), ESI-MS: calculated monoisotopic mass for C_49_H_68_F_7_N_9_O_10_: 1075.50; found: 1076.41 *m*/*z* [M+H]^+^.

### 2.2. Preparation of Radioconjugates

#### 2.2.1. ^68^Ga Radiolabeling

For the purpose of specific activity evaluation, different concentrations of APT-Oct-DOTA conjugate (10–25 nmol) dissolved in 410 µL of 0.2 M acetate buffer (pH 4.5) were incubated with a 50 MBq (300 µL) of [^68^Ga]GaCl_3_ in 0.1 M HCl from the ^68^Ge/^68^Ga generator at 95 °C for 10 or 30 min. After this time, each sample was instantly analyzed by HPLC for radiochemical yield (RCY) determination. For the purpose of physiochemical evaluation, each radioconjugate was obtained analogously with 3 MBq/nmol specific activity, purified by HPLC and evaporated.

#### 2.2.2. ^177^Lu Radiolabeling

For the purpose of specific activity evaluation, different amounts of APT-Oct-DOTA conjugate (0.5–5 nmol) dissolved in 200 µL of 0.2 M acetate buffer (pH 4.5) were incubated with a 5 MBq (5.2–6.7 µL) of [^177^Lu]LuCl_3_ in 0.04 M HCl at 95 °C for 10 or 60 min. After this time, each sample was instantly analyzed by HPLC for RCY determination. For the purpose of physiochemical evaluation, each radioconjugate was obtained analogously with 5 MBq/nmol specific activity, purified by HPLC and evaporated. However, for the purpose of biological evaluation all radioconjugates were obtained with 5 MBq/nmol specific activity in diluted 0.02 M acetate buffer (pH 4.5) and directly applied for cellular binding studies without any purification.

#### 2.2.3. Preparation of Non-Radioactive References

The non-radioactive Ga/Lu references were obtained by reaction of 220 µL of a 20 mM GaCl_3_ or 80 µL of a 20 mM LuCl_3_ in 0.1 M HCl with 100 nmol of the selected DOTA-conjugate in 300 µL of a 0.2 M acetate buffer (pH = 4.5) for 10 min at 95 °C. Subsequently, reaction products were purified by the HPLC method, lyophilized, and verified by mass spectrometry. Complexation yields for all conjugates were above 97%.

ESI-MS: calculated monoisotopic mass for **Ga-DOTA-Hex-APT,** C_45_H_57_F_7_N_9_O_10_Ga: 1085.34 and 1087.34; found: 1086.54 and 1088.53 *m*/*z* [M+H]^+^;

ESI-MS: calculated monoisotopic mass for **Ga-DOTA-Oct-APT,** C_47_H_61_F_7_N_9_O_10_Ga: 1113.37 and 1115.37; found: 1114.57 and 1116.57 *m*/*z* [M+H]^+^;

ESI-MS: calculated monoisotopic mass for **Ga-DOTA-Dec-APT,** C_49_H_65_F_7_N_9_O_10_Ga: 1141.40 and 1143.40; found: 1142.54 and 1144.59 *m*/*z* [M+H]^+^;

ESI-MS: calculated monoisotopic mass for **Lu-DOTA-Hex-APT,** C_45_H_57_F_7_N_9_O_10_Lu: 1191.35; found: 1192.54 *m*/*z* [M+H]^+^;

ESI-MS: calculated monoisotopic mass for **Lu-DOTA-Oct-APT,** C_47_H_61_F_7_N_9_O_10_Lu: 1219.38; found: 1220.53 *m*/*z* [M+H]^+^;

ESI-MS: calculated monoisotopic mass for **Lu-DOTA-Dec-APT,** C_49_H_65_F_7_N_9_O_10_Lu: 1247.42; found: 1248.59 *m*/*z* [M+H]^+^.

### 2.3. Physiochemical Evaluation of Radioconjugates

#### 2.3.1. Stability Study

A solution of purified ^177^Lu-radioconjugate (around 2.5 MBq) in 50 µL of 0.1M DPBS buffer (pH 7.40) was added into 450 µL of human serum and incubated at 37 °C for 7 days. After 1 day and at the end of incubation, the 200 µL of the incubated mixture was added into 400 µL of ethanol, vigorously stirred to precipitate serum proteins, and centrifuged at 13,500 rpm for 5 min to separate the supernatant, which was utilized for HPLC analysis with gamma measurement.

#### 2.3.2. Lipophilicity Study

A solution of purified ^177^Lu-radioconjugate (around 1 MBq) or ^68^Ga-radioconjugate (around 10 MBq) in 100 µL of 0.1M DPBS buffer was added into of 900 µL of 0.1 M DPBS buffer and 1000 µL of *N*-octanol (saturated with each other), vigorously stirred and centrifuged at 13,500 rpm for 5 min. Then, similar aliquots of both separated phases were taken for radioactivity measurement using a well-type NaI(Tl) detector. The lipophilicity values were expressed as the decimal logarithm of the distribution coefficient, D, which was calculated as a ratio of the radioconjugate radioactivity in the organic phase to that in the aqueous phase. Each experiment was performed in triplicate and the obtained values averaged. In parallel, the aqueous phases were analyzed by HPLC to confirm the stability of the studied radioconjugate during the experiment time.

### 2.4. Cell Culture

The Chinese hamster ovary CHO-K1 cell line with a stable overexpression of the human NK1 receptor (hNK1R-CHO cells) were obtained as a gift from Dr. Attila Keresztes and Dr. John M. Streicher [24]. Cells were grown in Ham’s F12 medium (Biological Industries, Beit HaEmek, Israel) supplemented with 10% Foetal Bovine Serum (Biological Industries, Beit HaEmek, Israel) and 400 μg/mL G418 (Capricorn Scientific, Ebsdorfergrund, Germany), incubated in a humidified atmosphere with 5% CO_2_ at 37 °C.

After reaching almost total confluence, the cells were washed with Dulbecco’s PBS (Biological Industries, Beit HaEmek, Israel), detached from the flasks using 0.05% trypsin-EDTA (Biological Industries, Beit HaEmek, Israel) at 37 °C, then diluted with medium and centrifuged for 10 min at 200× *g*. Obtained pellets were resuspended in the medium for the manual calculation of the cell quantity using Trypan Blue Solution (Biological Industries, Beit HaEmek, Israel) contrast.

### 2.5. Binding Affinity Determination

The saturation binding studies were performed according to the procedure published previously [4] for all ^177^Lu-radioconjugates obtained according to the common labeling protocol given above with the same specific activity of 5 MBq/nmol. In brief, 10^5^ cells per well were seeded into 24-well plates and incubated 24 h before the experiment. Just before the assay, the cells were washed with 37 °C Dulbecco’s PBS and then incubated with different concentrations of an analyzed radioconjugate (0.2–50 nM at a final volume) in the medium, with or without a 1000-fold molar excess of blocker (aprepitant or [Thi^8^,Met(O_2_)^11^]SP) in relation to the highest radioconjugate concentration applied for 60 min at 37 °C. After incubation time, the assay medium was collected into plastic tubes for radioactivity measurement and the cells were washed twice with ice-cold Dulbecco’s PBS. Subsequently, the cells were lysed using 1 M NaOH and collected into plastic tubes also for radioactivity measurement.

The radioactivity of collected samples were measured using a Wizard2 2-Detector Gamma Counter (PerkinElmer, Waltham, MA, USA). The study data came from three independent experiments done in duplicate. The Kd and BMAX with standard deviations (SD), were calculated using the “One site-total and nonspecific binding” nonlinear regression curve fit (GraphPad Prism 8, GraphPad, San Diego, CA, USA).

### 2.6. Docking

The complexes of the studied conjugates with the neurokinin 1 receptor were prepared in the following manner. The aprepitant structure (in neutral form) as found in the complex with the NK1R (PDB accession code: 6HLO [23]) was manually expanded by attaching linkers and DOTA moiety to the triazolinone ring. The geometry of DOTA-fragment was adjusted to mimic the geometry of DOTA in the HODCEI entry [25] of The Cambridge Structural Database [26]. This structure is one of the DOTA-Phe-NH_2_ complex with Y^3+^. In our modelling, DOTA carboxylate arms were protonated and frozen in the starting conformation in order to emulate the interactions of DOTA with the cation. Such an approach is hoped to give a rough approximation of the DOTA steric influence on the binding of the conjugates even though properly scaled and validated parameters for modelling and scoring of the complexes with the cations of interest are lacking.

For each compound, several different starting orientations were considered. Such initial complexes were subjected to local search docking in AutoDock 4.2.6 (Scripps Research Institute, La Jolla, CA, USA) [27].

The receptor structure for docking was a refined one (as provided by the GPCRdb service [28]). In this model, the mutated residues have been replaced with the native ones and the side chains missing in the original PDB structure have been added. The structure was pre-processed in AutoDock Tools [27]. The grids were calculated with AutoGrid 4 [27]. The docking box size was 34.5 Å × 34.5 Å × 41.25 Å.

All receptor residues were rigid. The ligands’ torsional freedom was allowed, except for the DOTA-fragment, that was frozen in the initial geometry. The docking procedure was the local search with the following parameters: 300 individuals in a population, 500 iterations of the Solis–Wets local search, local search space (*sw_rho* parameter) set to 100.0, and 1000 local search runs. The structures resulting from the local search were clustered and the representative models of the lowest scored (on average) cluster were taken for further analysis.

For qualitative assessment of the binding energy, both the lowest and the mean energy of the clusters were collected.

Molecular graphics were prepared in the Open-Source PyMOL [29] and in the Biovia Discovery Studio Visualizer [30].

## 3. Results and Discussion

### 3.1. Syntheses and Characterization of Aprepitant-Based Radioconjugates

#### 3.1.1. Syntheses of Aprepitant Derivative Conjugates

First, we have modified the structure of aprepitant in order to introduce a primary amino group following strategy similar to the one reported previously [1] with the aim of obtaining three new aprepitant-alkylamine derivatives (Figure 1). Guided by the fact that, the previously obtained alkylamine derivatives were the most promising, we synthesized further homologues in this series with longer aliphatic chains. Afterwards, we have performed the coupling reaction with DOTA-NHS ester to obtain desired aprepitant conjugates. The choice of only one radionuclide chelator was dictated by stability issues evaluated previously [1].

#### 3.1.2. Preparation of Radioconjugates and Labeling Optimization

To evaluate the labeling procedure of the obtained aprepitant-DOTA conjugates, 50 MBq of [^68^Ga]GaCl_3_ or 5 MBq of [^177^Lu]LuCl_3_ were incubated with different amounts (10–25 or 0.5–5 nmol, respectively) of APT-Oct-DOTA conjugate at 95 °C. The labeling RCYs determined at two time points using HPLC with gamma detection are presented in Table 1. The labeling of the APT-Hex-DOTA and APT-Dec-DOTA conjugates were performed under the conditions corresponding to the highest specific activity determined, i.e., 3 MBq/nmol for ^68^Ga and 5 MBq/nmol for ^177^Lu. Radiochromatograms of these labeling reactions of the newly obtained aprepitant radioconjugates are presented in Figure 1. All investigated aprepitant conjugates showed an excellent ability to complex metal radioisotopes with a satisfactorily high specific activity for further in vitro evaluation.

At the same time, syntheses of the non-radioactive reference compounds using stable gallium and lutetium were performed to verify the reliability of ^68^Ga and ^177^Lu-radioconjugate preparation. These syntheses were carried out in an analogous manner to those with radioactive metals, with the only difference being the use of an excess of metals in their concentrated solutions. The obtained non-radioactive references were purified using HPLC and followed by the characterization using mass spectrometry. The comparison of HPLC retention times (t_R_) of corresponding radioactive and stable metal aprepitant conjugates is presented in Table 2. For all pairs of radioconjugate and its non-radioactive reference, the obtained values are compatible (within ±0.1 min), which corroborates the identity of the investigated radioconjugates.

### 3.2. Physiochemical Evaluation of Radioconjugates

#### 3.2.1. Stability Study

The ^177^Lu-radioconjugates, purified using the HPLC method and evaporated, were examined as to their stability in human serum (HS). For this purpose, each radioconjugate was mixed with HS and incubated at 37 °C for period of 7 days. At both 1-day and 7-day time points, samples of the radioconjugate mixture were analyzed using the HPLC method for the assessment of the radioconjugate stability. All three novel ^177^Lu-radioconjugates corroborated their full stability in HS, the same as we reported previously [1].

#### 3.2.2. Lipophilicity Study

All six ^68^Ga/^177^Lu-radioconjugates (also purified using the HPLC method and evaporated) were evaluated in terms of their lipophilicity. For each radioconjugate was determined the distribution coefficient (D) between the organic and aqueous phases in the *N*-octanol/DPBS (pH 7.40) system. Simultaneously, the stability of investigated radioconjugate during the experiment was verified through the HPLC analysis of the aqueous phase. The lipophilicity values were established as the logarithm of the distribution coefficient acquired in three independent experiments done in duplicates, and are listed in Table 3.

For the newly obtained radiolabeled aprepitant derivatives, an increase in radioconjugate lipophilicity is clearly visible in relation to the previously obtained radioconjugates of aprepitant derivatives with an aliphatic or an acetamide linkers. Thus, an upward trend in the value of lipophilicity for a series of homologues containing an aliphatic linker was revealed, as intuitively expected.

Moreover, exactly the same manner as in the previous studies, novel radioconjugates containing ^177^Lu were more lipophilic than those with ^68^Ga; the differences between the radioconjugates obtained from the same precursor being around 0.3–0.65 logD units. Speaking in more detail, it was the first four conjugates containing shorter aliphatic linkers that showed similar differences in lipophilicity between the corresponding ^68^Ga and ^177^Lu labeling products (0.5–0.65 logD unit), while for APT-Oct-DOTA and APT-Dec-DOTA derived radioconjugates these differences were twice as low (0.32–0.37 logD unit). For the two pairs of the most lipophilic radioconjugates, namely **[^68^Ga]Ga/[^177^Lu]Lu-DOTA-Oct-APT** (logD = 1.32 and 1.64) and **[^68^Ga]Ga/[^177^Lu]Lu-DOTA-Dec-APT** (logD = 1.60 and 1.97), the effect of a long aliphatic linker noticeably eliminated the differences resulting from the different structure of the metal-DOTA complexes (cation coordination and the number of carboxylate group employed in metal chelation) and the significant hydrophilic contribution of the macrocyclic chelator to the resultant lipophilicity value of the radioconjugates. This fact may impact on diverse pharmacokinetic profiles among individual aprepitant-based radioconjugates, and the most lipophilic ones may behave similar to the parent drug. Nevertheless, it is difficult to unequivocally assess the ability of the obtained aprepitant radioconjugates to cross the blood–brain barrier. On the one hand, the lipophilicity of compounds is often expected to be a predicting factor for brain penetration, and the most optimal lipophilicity value is in the range of 2.0 to 3.5 [31]. The most lipophilic radioconjugates of aprepitant have slightly lower values than those, however, it should be borne in mind that oncological pathologies of the brain are often accompanied by the disruption of the blood–brain barrier [32].

### 3.3. Binding Affinity

The saturation binding studies using transfected hNK1R-CHO cells were performed to evaluate the binding characteristics of the ^177^Lu-radioconjugates of aprepitant derivatives in comparison to reference radioconjugate **[^177^Lu]Lu-DOTA-[Thi^8^,Met(O_2_)^11^]SP**. Obtained results confirm the hypothesis that the functionalization of the aprepitant molecule proposed by us allowed to obtain radiotracers with high binding affinity to the receptor of interest and sensitive to unmodified aprepitant blocking, likewise as is in the case of the reference [Thi^8^,Met(O_2_)^11^]SP. All determined K_d_ and B_MAX_ values for the investigated ^177^Lu-radioconjugates are presented in Table 4, supplemented with binding curves shown in Figure 2.

Binding affinity of all examined ^177^Lu-radioconjugates and reference radioconjugate **[^177^Lu]Lu-DOTA-[Thi^8^,Met(O_2_)^11^]SP** was relatively similar in low nanomolar range. The only distinctive K_d_ value was found for the **[^177^Lu]Lu-DOTA-Et-APT** radioconjugate (containing the shortest aliphatic linker, K_d_ = 18.9 nM), which was more than threefold worse than that of next in order, determined for **[^177^Lu]Lu-DOTA-Dec-APT** (K_d_ = 6.26 nM). The reference **[^177^Lu]Lu-DOTA-[Thi^8^,Met(O_2_)^11^]SP** showed values of K_d_ and B_MAX_ equal to 2.74 nM and 0.38 nM, respectively, with high specific binding at the level of about 85%.

The highest binding affinities were obtained for **[^177^Lu]Lu-DOTA-Hex-APT** (K_d_ = 1.56 nM), **[^177^Lu]Lu-DOTA-Bu-APT** (K_d_ = 1.66 nM), and **[^177^Lu]Lu-DOTA-Oct-APT** (K_d_ = 2.48 nM). These values were better than the one found for the reference radioconjugate. Then, K_d_ values similar to K_d_ value of the reference were found for **[^177^Lu]Lu-DOTA-NH-NH-Ac-APT** (K_d_ = 2.77 nM) and **[^177^Lu]Lu-DOTA-Pr-APT** (K_d_ = 2.92 nM). Thus, an interesting trend can be observed for radioconjugates with aliphatic linkers (Figure 2), since initially the affinity for the receptor increases with linker lengthening for radioconjugates with butyl and hexyl linker the affinity is optimal, and then it decreases with further elongation of the aliphatic linker in the molecule.

A phenomenon of particular importance is the fact that all eight aprepitant-based radioconjugates showed more than threefold higher binding capacity (B_MAX_ in range from 1.24 to 9.16 nM) than that of the reference SP-based radioconjugate (B_MAX_ = 0.38 nM). Moreover, this is in line with the observation made for a similar comparison of non-peptide L732,138-based radioconjugates to the same reference compound [4].

For all radioconjugates, we also determined the ratio of specific binding to total binding (expressed as a percentage, last column in Table 4) using the values corresponding to the radioconjugate concentration of 25 nM (at this concentration, the plateau generally begins in the specific binding curves, Figure 3). This parameter presents the trend that as the aliphatic linker length increases, the share of specific binding drops in favor of nonspecific binding. Furthermore, for the two most lipophilic radioconjugates, **[^177^Lu]Lu-DOTA-Oct-APT** and **[^177^Lu]Lu-DOTA-Dec-APT**, at the concentration of 25 nM, nonspecific binding exceeded specific binding. In terms of the percentage of specific binding, radioconjugates with the lowest lipophilicity look best, **[^177^Lu]Lu-DOTA-Et-APT** (above 90%), **[^177^Lu]Lu-DOTA-Et-Ac-APT** (slightly below 90%) as well as **[^177^Lu]Lu-DOTA-NH-NH-Ac-APT** and **[^177^Lu]Lu-DOTA-Pr-APT** (both about 75%).

### 3.4. Molecular Modelling Study

In order to rationalize the obtained experimental affinities in terms of ligand-receptor interactions, the conjugates containing aliphatic linkers were modelled in complex with the human NK1R. The aprepitant structure (as found in the crystal with the receptor, PDB accession code: 6HLO [23]) was manually expanded with linker-DOTA fragments and the local search docking was performed in AutoDock 4.2.6 (Scripps Research Institute, La Jolla, CA, USA) [27]. The studied model ligands did not include Lu^3+^ cation, therefore their names are marked with the asterisk, e.g., ***DOTA-Et-APT**, etc.

With all modelled conjugates, the overall architecture of the binding mode is similar (Figure 4). The aprepitant core resides deep in the binding pocket, in a position slightly displaced with regard to the position of aprepitant in the 6HLO crystal structures [23] (Figure 4A, Appendix A). A major difference, compared to the small molecular parent, is the rotation of the triazolinone ring (at which the linker fragment is attached) that prevents interactions of this element with E193 and W184. The linker-DOTA fragments of the studied ligands extend towards the binding site outlet (Figure 4B,C). In shorter analogues, the linkers are rather extended and DOTA-moiety is placed close to the extracellular tip of transmembrane helix 5 (TM5) and the extracellular loop 2 (ECL2). On the contrary, in octyl- and decyl-based analogues, the linkers are bent and DOTA-fragment approaches ECL1, TM3 and TM2 (Figure 4C).

The binding interactions are discussed below for the ***DOTA-Hex-APT** ligand, which corresponds to **[^177^Lu]Lu-DOTA-Hex-APT** for which the lowest K_d_ value was found (Figure 5 and Appendix A). 3,5-bis-trifluoromethylphenyl ring of the ligand forms numerous hydrophobic interactions with M81, N89, P112, V116, I204, W261, M291, A294 and M295. The morpholine ring is hanged between F268 and Q165 side chains. The latter provides further stabilization of the binding mode by hydrogen bonding with the ether oxygen of the ligand. The *p*-fluorophenyl ring stacks with the side chain of F264. Additional contacts of this fragment are to V200, T201 and H265. The triazolinone ring *π*-stacks with H197. The hexyl linker runs along TM5, contacting K190 and E193. The chelating moiety lays against ECL2, forming interactions with, i.e., M174, M181, E183 and E186.

The interactions of the remaining analogues in the part common to all of them (‘aprepitant core’) are generally similar to these described for ***DOTA-Hex-APT**, although some details may differ for particular compounds (see Appendix A for detailed schemes of interactions). The presence of the linker and DOTA fragments has little impact on the position of the ‘aprepitant core’, and it is from this very part of the considered molecules that most of the binding strength is likely to come. This finding fits in qualitative terms to the experimental affinities reported in this contribution.

Quantitative prediction of affinities from docking is given in Table 5. Unfortunately, neither the lowest nor the mean predicted binding energies of the best scored cluster correlate with the experimental affinities. This could be associated with artificial overpenalization of conformational freedom for the analogues with long flexible linkers.

The issue of conformational freedom is undoubtedly a limitation of the current modelling study. For the longest analogues, it is very likely that more than one binding mode is realized (with respect to the DOTA-linker fragment). Indeed, the performed docking shows that there are many different (but closely lying in energy) docked clusters (not reported) for the longest analogues. Moreover, in our recent contribution [4], molecular dynamics simulations of small molecular NK1R ligands with long (14 atoms) linkers found significant mobility of the pendant fragment despite relative stability of the deep-bound core. This problem is going to be addressed in the further modelling study. Nevertheless, the binding modes reported herein are going to guide further design work in the field of NK1R-targeted radiopharmaceuticals.

## 4. Conclusions

The contribution presented herein describes the evaluation of three novel aprepitant-based DOTA conjugates, effectively labeled with ^68^Ga and ^177^Lu with high RCY and specific activity. These developed radioconjugates were characterized by high lipophilicity and full stability in human serum. In vitro investigation on binding characteristics towards NK1R of novel ^177^Lu-radioconjugates showed their high receptor affinity (K_d_ = 1.56–6.26 nM), but concomitantly revealed significant non-specificity (up to 60% of specific binding).

At the same time, most of previously reported radioconjugates based on less lipophilic aprepitant derivatives presented much the same high receptor affinity (K_d_ = 1.66–5.17 nM) like the reference **[^177^Lu]Lu-DOTA-[Thi^8^,Met(O_2_)^11^]SP** (K_d_ = 2.74 nM), however, with the binding capacity more than three times higher (B_MAX_ = 1.24–5.29 nM) in relation to the SP-based reference radioconjugate (B_MAX_ = 0.38 nM). Furthermore, it was the radioconjugates with the lowest lipophilicity that had the highest specific binding rate (75% and above), at a level comparable to that of the reference radioconjugate.

Assessing the overall properties of all eight radiopharmaceuticals developed, in particular the lipophilicity parameter and the tendency to decrease the share of specific binding with an increase in the length of the aliphatic linker, the most promising radiopharmaceutical seem to be **[^177^Lu]Lu-DOTA-Pr-APT** radioconjugate, containing three CH_2_ groups in the aliphatic linker between the radionuclide complex and an aprepitant molecule.

In conclusion, our efforts confirmed the assumptions about the effectiveness of aprepitant in the role of a NK1R targeting vector, in particular referring to ^177^Lu-labeled Substance P derivative currently used in clinical therapy. An application of this NK1R antagonist in nuclear medicine may initiate a potential alternative approach to NK1R-positive cancer imaging following intravenous administration. The reported results provide the encouraging perspectives for the future first in vivo trials.

## Data Availability

Not applicable.

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
