# Peer review of "In Vitro Biological Evaluation of Aprepitant Based 177Lu-Radioconjugates"

_pharmaceutics, 2022, doi:10.3390/pharmaceutics14030607_

Round 1

Reviewer 1 Report

The manuscript by Paweł K. Halik et al. submitted to Pharmaceutics described the biological evaluation of some 177Lu-radioconjugates of aprepitant. The in vitro experiments, labeling optimization, radioconjugates characterization and docking modeling were also conducted. The manuscript seems to be of broad interest for chemist and biologist working in the area and should be published after major revision.

Comments:

  • Because the title of the manuscript is “the biological evaluation of some 177Lu-radioconjugates of aprepitant”, the authors should conduct the biodistribution studies and blocking studies in mice bearing tumor of the 177Lu-radioconjugates to verify their specific abilities to target tumor.
  • The authors should also conduct the SPECT imaging studies and blocking studies in mice bearing tumor of the 177Lu-radioconjugates.
  • On page 3, 80°C should be 80 °C.

Author Response

Ad Manuscript ID: pharmaceutics-1611282

Title: Biological Evaluation of Aprepitant based 177Lu-Radioconjugates.

First of all, we would like to thank the Editorial Office and Reviewers for their effort and all constructive comments. We carefully analyzed all the issues raised by the Reviewers, so we made appropriate corrections to the manuscript. Please, find below answers to the Reviewers' comments in the received order:

Responses to the comments of Reviewer 1:

Comments and Suggestions for Authors:

The manuscript by PaweÅ‚ K. Halik et al. submitted to Pharmaceutics described the biological evaluation of some 177Lu-radioconjugates of aprepitant. The in vitro experiments, labeling optimization, radioconjugates characterization and docking modeling were also conducted. The manuscript seems to be of broad interest for chemist and biologist working in the area and should be published after major revision.

Thank you for your positive overall evaluation of our manuscript. Nevertheless, it seems to us that the article does not generally require major revision. Please read our position on your suggestions and we hope that the Reviewer will accept our explanations.

Comments:

  • Because the title of the manuscript is “the biological evaluation of some 177Lu-radioconjugates of aprepitant”, the authors should conduct the biodistribution studies and blocking studies in mice bearing tumor of the 177Lu-radioconjugates to verify their specific abilities to target tumor.
  • The authors should also conduct the SPECT imaging studies and blocking studies in mice bearing tumor of the 177Lu-radioconjugates.

According to the authors' intention, the research presented in the article was to concern all significant physicochemical and biological tests of potential radiopharmaceuticals based on a novel modified aprepitant molecules that could be carried out in vitro. Such a set of tests in vitro is required before conducting research on living organisms. Of course, further in vivo research on animal models of new potential radiopharmaceuticals is necessary and of fundamental scientific importance. For the best radioconjugates selected on the basis of the presented results of in vitro tests described in the submitted article, we plan to conduct the multi-organ biodistribution study on healthy mice and animal glioblastoma xenograft models, but this will be the subject of a new scientific report already.

In order to clarify the title of our article and its content, we introduced into the title of the manuscript the information that all tests, including biological evaluation, were performed in vitro.

  • On page 3, 80°C should be 80 °C.

Naturally, the notation has been changed as suggested.

Regardless of the Reviewers' requests, we have decided also to add a description of the 1H-NMR analyzes for the newly obtained aprepitant derivatives in order to improve the comprehensiveness of this report.

Reviewer 2 Report

The research article “Biological Evaluation of Aprepitant based 177Lu-Radioconjugates” describes the synthesis and evaluation of several Aprepitant compounds to target the NK1 receptor. This study nicely builds on a previous study published by the same research group. The paper uses in vitro assays to find the optimal length of a spacer between the DOTA moiety and aprepitant binding moiety. The in vitro data were complemented with computational docking studies. The small molecule approach to circumvent the brain penetration issues of protein based radiopharmaceuticals is an concept which is worthwhile investigating. The paper is nicely written and tells a coherent story. Due to the reasons mentioned above, I believe this is an interesting paper. The authors should however address a few minor remarks.

Minor remarks:

The article itself mentions potential applications of the Aprepiant based radiopharmaceuticals in several cancers including glioma. Therefore, brain penetration is crucial. The lipophilic nature of compounds is often expected to be a predicting factor for brain penetration. It could be interesting to include this discussion in the Lipophilicity section.

Page 3 Line 130: “…and supplemented with a Thriethylamine…” remove the a

Page 3 line 134: “…verified by MS analysis confirmation…”: wrong sentence. Remove confirmation or rephrase verified

Page 5 line 203: “The lipophilicity values was…” subject is plural

Page 8 Table 2: TR determined based on multiple measurements? Please add SD.

Page 13 Table 5: add SD to Mean

Author Response

Ad Manuscript ID: pharmaceutics-1611282

Title: Biological Evaluation of Aprepitant based 177Lu-Radioconjugates.

First of all, we would like to thank the Editorial Office and Reviewers for their effort and all constructive comments. We carefully analyzed all the issues raised by the Reviewers, so we made appropriate corrections to the manuscript. Please, find below answers to the Reviewers' comments in the received order:

Responses to the comments of Reviewer 2:

Comments and Suggestions for Authors:

The research article “Biological Evaluation of Aprepitant based 177Lu-Radioconjugates” describes the synthesis and evaluation of several Aprepitant compounds to target the NK1 receptor. This study nicely builds on a previous study published by the same research group. The paper uses in vitro assays to find the optimal length of a spacer between the DOTA moiety and aprepitant binding moiety. The in vitro data were complemented with computational docking studies. The small molecule approach to circumvent the brain penetration issues of protein based radiopharmaceuticals is an concept which is worthwhile investigating. The paper is nicely written and tells a coherent story. Due to the reasons mentioned above, I believe this is an interesting paper. The authors should however address a few minor remarks.

Thank you for your appreciation of our efforts as well as carefully prepared comments and observations. Please find our answers to the provided remarks.

Minor remarks:

The article itself mentions potential applications of the Aprepiant based radiopharmaceuticals in several cancers including glioma. Therefore, brain penetration is crucial. The lipophilic nature of compounds is often expected to be a predicting factor for brain penetration. It could be interesting to include this discussion in the Lipophilicity section.

Thank you for this recommendation. We briefly broadened that issue in the Lipophilicity Study section (3.2.2), indicating how the results we obtained relate to the optimal lipophilicity values for radiotracers capable of crossing the blood-brain barrier.

  • Page 3 Line 130: “…and supplemented with a Thriethylamine…” remove the a
  • Page 3 line 134: “…verified by MS analysis confirmation…”: wrong sentence. Remove confirmation or rephrase verified
  • Page 5 line 203: “The lipophilicity values was…” subject is plural

Indeed, some small mistakes occur in the text, thereby the entire manuscript was carefully proofread and spelling errors are corrected.

  • Page 8 Table 2: TRdetermined based on multiple measurements? Please add SD.

Unfortunately, the tR values for reference compounds with stable metals were determined on the basis of a single HPLC analysis. For radioconjugates, we also performed one HPLC analysis to determine this specific value, however, as a result of further experiments (stability evaluation, lipophilicity, labeling optimization), we repeatedly obtained the same retention time values for each radioconjugate.

  • Page 13 Table 5: add SD to Mean

This is an important point, the averaged results given in the table 5 have been now supplemented with the values of the standard deviation.

Regardless of the Reviewers' requests, we have decided also to add a description of the 1H-NMR analyzes for the newly obtained aprepitant derivatives in order to improve the comprehensiveness of this report.

Round 2

Reviewer 1 Report

The authors's explanation can be accepted.